# ATP depletion in anthrax edema toxin pathogenesis

Jie Liu[1,2◉], Qing Cao[1,3◉], Michael Ewing[1], Zehua Zuo[1], Jason R. Kennerdell[1], Toren Finkel[1,2], Stephen H. Leppla[4], Shihui Liu◉[1,5]*

1 Aging Institute of University of Pittsburgh and University of Pittsburgh Medical Center, Pittsburgh, Pennsylvania, United States of America, 2 Division of Cardiology, Department of Medicine, University of Pittsburgh School of Medicine, Pittsburgh, Pennsylvania, United States of America, 3 PhD program, Department of Environmental and Occupational Health, University of Pittsburgh School of Public Health, Pittsburgh, Pennsylvania, United States of America, 4 Microbial Pathogenesis Section, Laboratory of Parasitic Diseases, National Institute of Allergy and Infectious Diseases, National Institutes of Health, Bethesda, Maryland, United States of America, 5 Division of Infectious Diseases, Department of Medicine, University of Pittsburgh School of Medicine, Pittsburgh, Pennsylvania, United States of America,

◉ These authors contributed equally to this work.
* SHL176@pitt.edu

## Abstract

Anthrax lethal toxin (LT) and edema toxin (ET) are two of the major virulence factors of *Bacillus anthracis*, the causative pathogen of anthrax disease. While the roles of LT in anthrax pathogenesis have been extensively studied, the pathogenic mechanism of ET remains poorly understood. ET is a calmodulin-dependent adenylate cyclase that elevates intracellular cAMP by converting ATP to cAMP. Thus, it was postulated that the ET-induced *in vivo* toxicity is mediated by certain cAMP-dependent events. However, mechanisms linking cAMP elevation and ET-induced damage have not been established. Cholera toxin is another bacterial toxin that increases cAMP. This toxin is known to cause severe intestinal fluid secretion and dehydration by cAMP-mediated activation of protein kinase A (PKA), which in turn activates cystic fibrosis transmembrane conductance regulator (CFTR). The cAMP-activated PKA phosphorylation of CFTR on the surface of intestinal epithelial cells leads to an efflux of chloride ions accompanied by secretion of $H_2O$ into the intestinal lumen, causing rapid fluid loss, severe diarrhea and dehydration. Due to similar *in vivo* effects, it was generally believed that ET and cholera toxin would exhibit a similar pathogenic mechanism. Surprisingly, in this work, we found that cAMP-mediated PKA/CFTR activation is not essential for ET to exert its *in vivo* toxicity. Instead, our data suggest that ET-induced ATP depletion may play an important role in the toxin's pathogenesis.

## Author summary

Anthrax ET is a calmodulin-dependent adenylate cyclase that elevates intracellular cAMP by converting ATP to cAMP. Thus, it was postulated that the ET-induced *in vivo* toxicity is mediated by certain cAMP-dependent events. PKA and Epac are the two major intracellular targets activated by cAMP. Surprisingly, in this work, we found that cAMP-mediated PKA/CFTR activation as well as Epac activation are not essential for ET to

**Data availability statement:** All data supporting the findings of this study are available within the paper and its Supplementary Information.

**Funding:** This research was supported by the institutional seed fund from the Aging Institute of University of Pittsburgh School of Medicine (S.L.) and the grant (R01AI170574) (S.L.) from the National Institute of Allergy and Infectious Diseases (NIAID), NIH, and in part by the Intramural Program of the NIAID (S.H.L), NIH. The funders had no role in study design, data collection and analysis, decision to publish, or preparation of the manuscript.

**Competing interests:** The authors have declared that no competing interests exist.

exert its *in vivo* toxicity. Instead, our data suggest that ET-induced ATP depletion may play an important role in the toxin's pathogenesis. Our results suggest that ATP depletion might also be an important mechanism underlying pathogenesis of other adenylate cyclase toxins.

## Introduction

*Bacillus anthracis* is a Gram-positive, rod-shaped, spore-forming bacterium and is the causative pathogen of anthrax disease. The *B. anthracis* spore, the infectious form of the pathogen, has long been considered as a potential warfare agent and was a top bioterrorism concern even before the 2001 anthrax attacks in the USA [1]. *B. anthracis* contains two large extra-chromosomal pathogenic plasmids, pXO1 (182 kb) and pXO2 (96 kb) [2,3]. Plasmid pXO2 encodes proteins that synthesize the unique poly-D-γ-glutamic acid capsule, which disguises *B. anthracis* from host immune surveillance and confers resistance to phagocytosis by host macrophages [4]. Plasmid pXO1 encodes the three components of anthrax exotoxins: the cellular binding/delivering moiety– protective antigen (PA, 83 kDa), and two enzymatic effector proteins– lethal factor (LF, 89 kDa), and edema factor (EF, 90 kDa) [4]. To target host cells, PA binds to cell surface anthrax toxin receptors CMG2 (capillary morphogenesis protein-2, the major receptor) and TEM8 (tumor endothelium marker-8), leading to proteolytic activation of PA by a cell surface furin protease, yielding the active PA oligomer (heptamer and/or octamer) [4,5]. The PA oligomer then binds and translocates LF and/or EF into the cytosol of target cells to exert their cytotoxic effects. Thus, PA and LF pair to form anthrax lethal toxin (LT), while PA plus EF form anthrax edema toxin (ET), resulting in the two major virulence factors of *B. anthracis*. LF (the catalytic component of LT) is a zinc-dependent metalloproteinase that cleaves the mitogen-activated protein kinase kinases (MEKs) 1-4 and 6 [6–9], resulting in inactivation of the three key mitogen-activated protein kinase pathways: ERK (through MEK1/2), p38 (through MEK3/6), and JNK (Jun N-terminus kinase) (through MEK4), leading to LT-induced lethality of experimental mice [4,5,10,11]. Since many human cancers rely on ERK pathway for tumorigenesis and growth, LT has been extensively reengineered to selectively target those tumors [12–16].

ET is another major virulence factor of *B. anthracis*. This is evidenced by the 10-fold reduction in virulence of *B. anthracis* when the gene encoding EF (the catalytic component of ET) is deleted [17,18]. EF is a calmodulin-dependent adenylyl cyclase that converts ATP to cAMP, the classical second messenger [19]. Thus, it was postulated that ET-induced *in vivo* toxicity is mediated by certain cAMP-dependent events [4,5]. However, mechanisms linking cAMP elevation and ET-induced damage are poorly understood. cAMP has two major intracellular targets: protein kinase A (PKA) and the exchange protein activated by cAMP (Epac), which is a guanine nucleotide exchange factor/activator for the small G-proteins Rap1 and Rap2 [20–23]. Other bacterial toxins that increase cAMP include cholera toxin, secreted by *Vibrio cholera*. Cholera toxin causes severe intestinal fluid secretion and dehydration by cAMP-mediated activation of PKA, which in turn activates cystic fibrosis transmembrane conductance regulator (CFTR) [24,25]. CFTR is an ATP-binding cassette transporter-class chloride ion channel found in epithelial cells of many organs, including the lung, liver, digestive tract, and skin. The cAMP-activated PKA phosphorylates CFTR on the surface of intestinal epithelial cells, leading to efflux of chloride ions accompanied by secretion of $H_2O$ into the intestinal lumen, causing rapid fluid loss and severe diarrhea and dehydration [25]. Due to the similarity of ET and cholera toxin in their ability to cause fluid loss, it was believed the two toxins had similar mechanisms of action. In this work, we sought to investigate whether PKA-mediated

CFTR activation is the cause of ET-induced edema and *in vivo* toxicity. Surprisingly, we found that cAMP-mediated PKA/CFTR activation is not essential for ET to exert its *in vivo* toxicity. Instead, our data suggest that ET-induced ATP depletion may play an important role in ET's pathogenesis.

## Results

### CMG2 receptor-dependent *in vivo* toxicity of anthrax edema toxin

The *in vivo* pathogenicity of ET has been carefully assessed since the highly purified EF protein was made available two decades ago [5,26–28]. In addition to the characteristic skin edema when injected subcutaneously, ET is also lethal when administered systemically. In fact, the minimal lethal doses of ET to mice (10-20 µg) are even lower than observed for LT (25-50 µg) [26,29]. Aligning well with this, here, we showed that administration of 20 µg ET (I.P.) resulted in a 60% mortality rate in challenged mice within two days, in a CMG2 receptor-dependent manner (Fig 1A). ET is an adenylate cyclase that quickly increases cellular cAMP levels in cultured cells [19]. To assess this effect *in vivo*, we measured the cAMP levels in plasma and in tissue lysates of the livers, kidneys, and hearts of mice 18 hours after ET administration. CMG2 receptor-dependent cAMP increases were detected in the livers but not in the kidneys (Fig 1B). The trend to increase cAMP in the hearts was also demonstrated. Surprisingly, significant increases (about 100-fold) of cAMP were also detected in the plasma, revealing that cellular cAMP can efficiently move out of the cells (Fig 1B). However, the mechanism and the significance of this cAMP translocation in ET pathogenesis remain unclear.

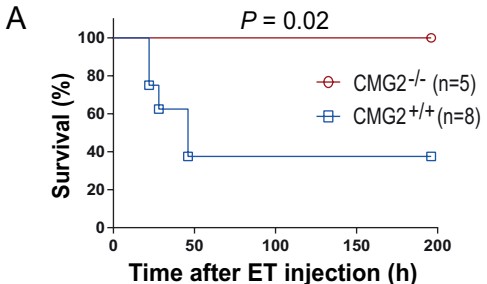

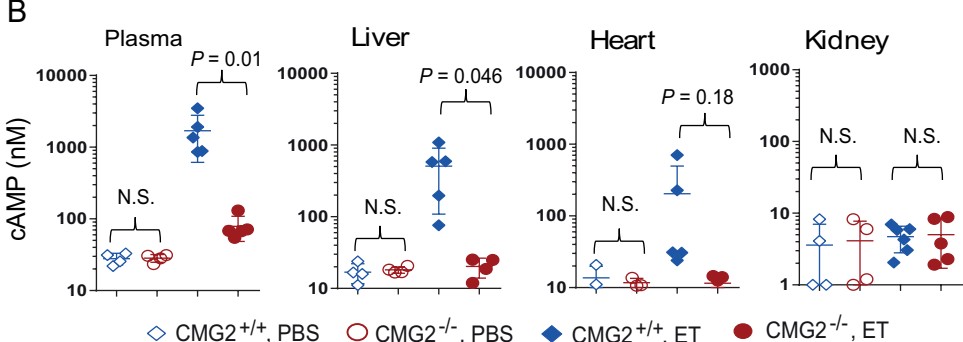

**Fig 1. CMG2 receptor-dependent *in vivo* toxicity of anthrax edema toxin. A**. CMG2-KO mice and their littermate WT control mice were challenged with 20 µg ET (I.P.), and monitored for signs of malaise and survival for 200 hours. Log-rank test for survival comparison. **B**. ET-induced cAMP increases in mice. Eighteen hours after challenge with 30 µg ET (I.V.), the mice (CMG2-KO and the controls) were euthanized, and blood, liver, heart, kidneys were collected and processed for measuring cAMP levels. Unpaired two-tailed Student's *t*-test.

## PKA- and CFTR-independent *in vivo* toxicity of anthrax edema toxin

The *in vivo* toxicity of ET was initially considered to be mediated by certain cAMP-activated pathological processes. cAMP has at least two distinct intracellular targets: PKA and Epac (Fig 2A) [20–23]. Epac is a guanine nucleotide exchange factor that activates the small G-proteins Rap1 and Rap2, which have positive roles in maintaining the tight junctions between epithelial cells [22,23,30,31]. Thus, activation of Epac by cAMP seems unlikely to mediate the ET's *in vivo* toxicity. Consistent with this hypothesis, we found that the Epac specific inhibitor ESI-09 could not protect mice from ET challenges (S1 Fig). PKA exists as an inactive tetramer consisting of two regulatory and two catalytic subunits. Binding of cAMP to the regulatory subunits causes their dissociation from the two catalytic monomers of PKA, which are then activated by ATP binding and are able to phosphorylate/activate many substrate proteins, including the transcription factor cAMP response element-binding protein (CREB) and the cystic fibrosis transmembrane conductance regulator (CFTR).

To examine whether ET-induced cAMP could activate PKA, we treated NIH 3T3 cells with ET, and found that phosphorylation (activation) of CREB was enhanced (Fig 2B). The PKA inhibitor H89 completely blocked this ET-induced CREB activation (phosphorylation) (Fig 2B). Further H89 could also completely block CREB activation by forskolin [32], another cAMP-increasing agent that increases cAMP through activating the endogenous adenylyl cyclase (Fig 2B). However, although H89 administration could efficiently diminish ET-induced activation of CREB in ET-challenged mice (Fig 2C), this PKA inhibitor could not provide survival benefit for the ET-challenged mice (Fig 2D).

To determine whether the activation of PKA or its downstream target CFTR is involved in ET-induced edema, we tested the effects of H89, as well as CFTRinh-172 [33], a specific CFTR inhibitor, on ET-induced edema using a mouse footpad edema model. ET could rapidly cause edematous lesions in this mouse footpad edema experiment. This pathology could only be marginally ameliorated by local pre-administration of either H89 or CFTRinh-172 (S2A and S2B Fig). However, the footpad edema caused by cholera toxin, another cAMP-increasing bacterial toxin [25], could be more pronouncedly reduced by H89 or CFTRinh-172 (S2C and S2D Fig).

CFTR-null (*CFTR*^{-/-}) mice die within a month after birth due to severe intestinal occlusion [34]. However, this phenotype of *CFTR*^{-/-} mice can be corrected when CFTR is specifically expressed in intestines through crossing with the intestinal-specific CFTR transgenic mice (*Fabp2-CFTR* mice) (CFTR is under the control of an intestinal-specific promotor rat fatty acid binding protein 2 (Fabp2) [35]. As such, *CFTR*^{-/-}/*Fabp2-CFTR* mice, which express CFTR only in intestines but not in other tissues, have normal longevity, allowing us to assess the potential role of CFTR in ET pathogenesis. To determine whether the ET-induced *in vivo* toxicity is CFTR-independent, we generated and challenged the *CFTR*^{-/-}/*Fabp2-CFTR* mice and their control mice with ET by intraperitoneal administration. We found that *CFTR*^{-/-}/*Fabp2-CFTR* mice and their WT control mice were equally susceptible to ET-induced lethality (Fig 2E). Similarly, systemic administration of CFTRinh-172 could not provide survival benefit for the mice challenged with lethal doses of ET (Fig 2F). Taken together, we conclude that cAMP-mediated activation of PKA, as well as its downstream target CFTR, which appear to be involved in ET-caused edema, are not required for the ET-induced lethality.

## Potential role of ATP depletion in ET pathogenesis

One characteristic cytotoxic effect of ET to cultured cells is its ability to rapidly induce morphological change including cell rounding (Fig 3A). This effect is also toxin receptor-dependent, since ET did not morphologically alter anthrax toxin receptor-deficient Chinese

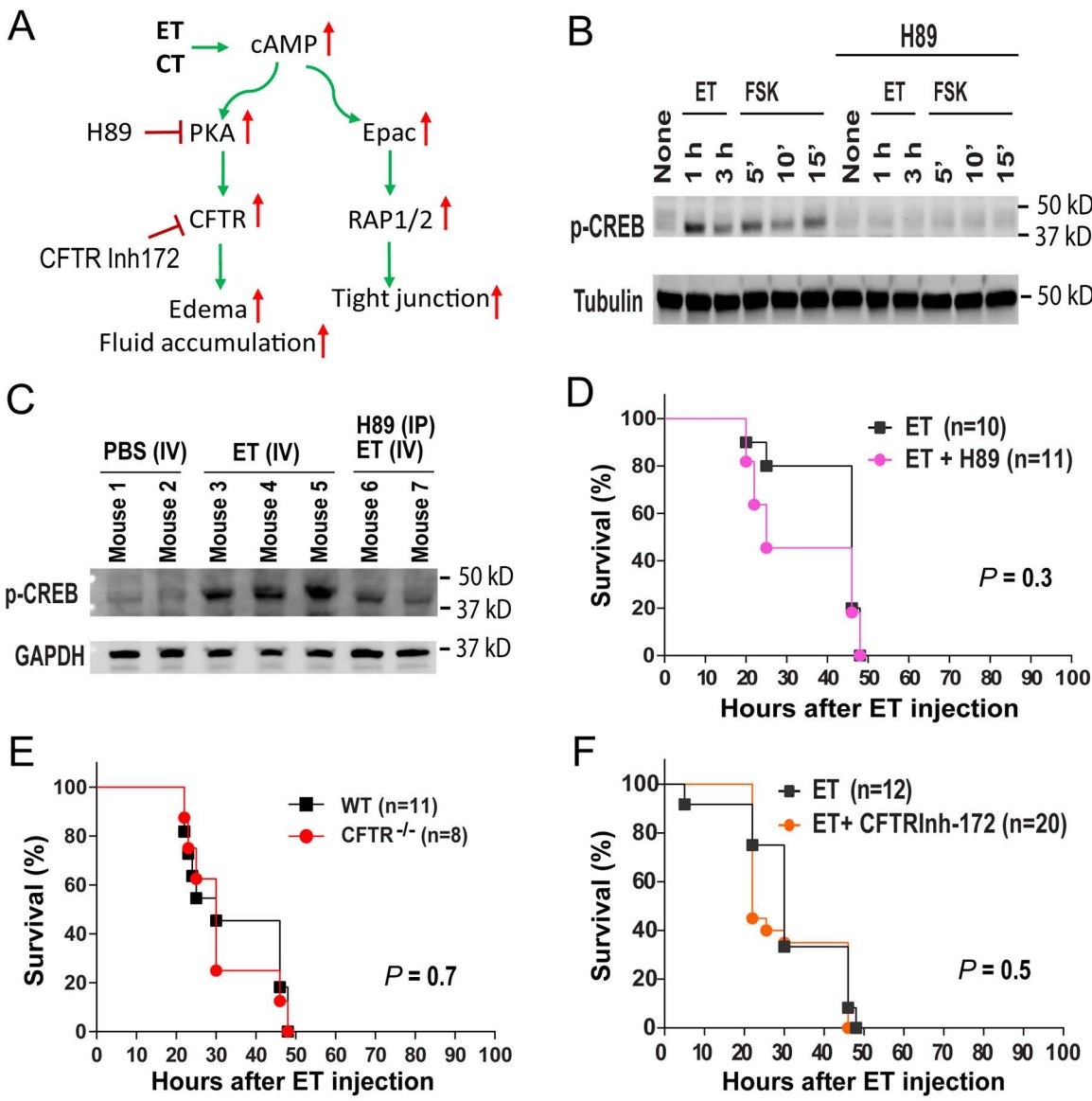

**Fig 2. PKA-independent *in vivo* toxicity of anthrax edema toxin. A.** cAMP-increasing bacterial toxins and cAMP-activated pathways. **B**. Activation of PKA by ET and forskolin. NIH3T3 cells were treated with ET (1 μg/mL) or FSK (forskolin, 10 μM) for various lengths of time as indicated with/without the presence of H89 (0.1 mM). The cell lysates were then analyzed by Western blotting using anti-phospho-CREB and anti-tubulin antibodies. Reprehensive of two independent experiments with similar results. **C**. Systemic administration of H89 effectively prevents ET-induced activation of CREB in the hearts of the ET-challenged mice. Mice were challenged with 30 μg ET (I.V.) or 30 μg ET (I.V.) plus 500 μg H89 (I.P.). Six 6 hours after administration, mice were euthanized and the heart tissue lysates prepared for Western blotting analysis using the antibodies as indicated (50 μg proteins loaded per lane). Of note, H89 could effectively diminish the ET-induced CREB activation. **D**. PKA inhibitor could not prevent ET-induced mortality. Mice were administered 30 μg ET (I.V.) or 500 μg H89 (I.P.) plus 30 μg ET (I.V.) and monitored for survival as indicated. **E**. CFTR-null (CFTR-KO (FABP-CFTR+)) mice and their WT control mice were challenged with 30 μg ET (I.V.) and monitored for survival. **F**. CFTR inhibitor could not prevent the ET-induced mortality. The mice were administered 30 μg ET (I.V.). 24 h and 1 h prior to ET injection, the mice were injected with PBS or 50 μg CFTRinh-172. (D-F) Log-rank test for survival comparison.

hamster ovary (CHO) PR230 cells, but could do so to PR230 cells following reconstitution with the CMG2 receptor (PR230 (CMG2)) (Fig 3A). Interestingly, along with these morphological changes, extracellular cAMP was noted to rapidly accumulate in the conditioned

medium of the ET-treated PR230 (CMG2) cells, but not in the conditioned medium of PR230 cells (Fig 3B). To begin to study the potential cytotoxic effect of extracellular cAMP, PR230 cells and PR230 (CMG2) cells grown on different cover slips were co-cultured in a same chamber and were treated with ET (Fig 3C). Although these cells were in the same conditioned medium, morphological change only occurred with the PR230 (CMG2) cells, and not with toxin receptor-deficient PR230 cells (Fig 3C), suggesting that the cell rounding effect of ET was not likely attributable to the accumulation of the extracellular cAMP.

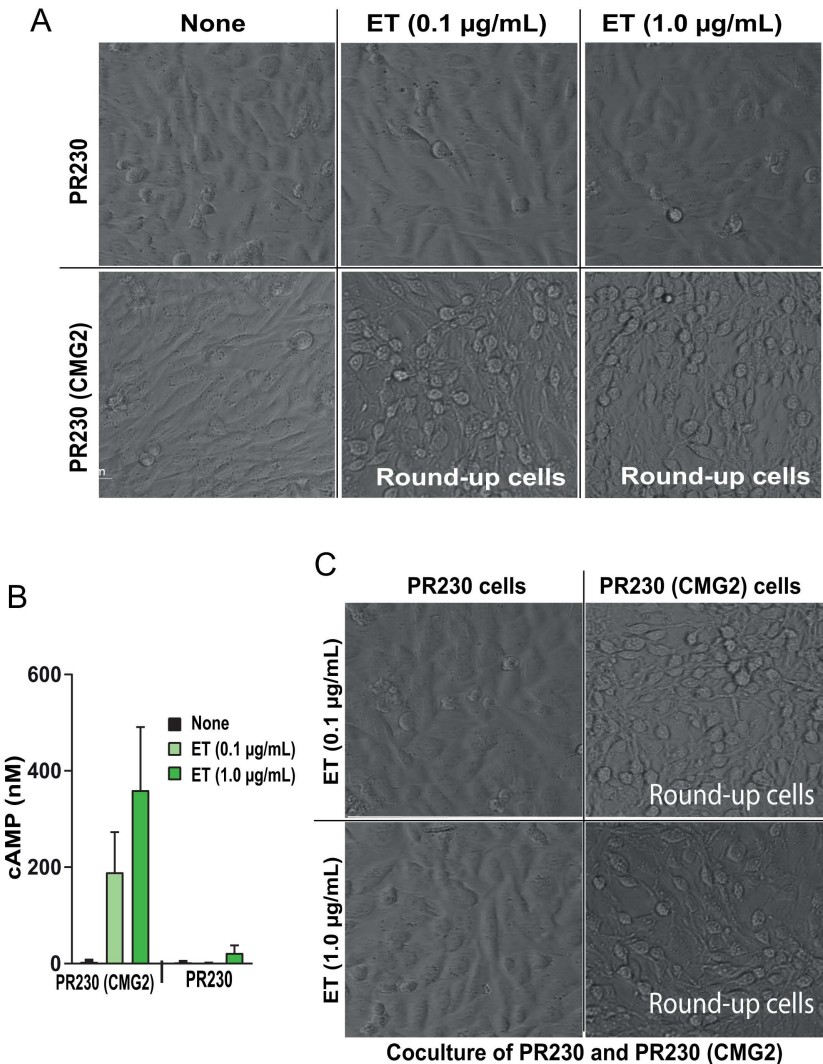

**Fig 3. The extracellular accumulation of cAMP is dispensable for the ET-induced cell rounding morphological change. A**, **B**. Toxin receptor-dependent cell rounding effect of ET. PR230 (a PA receptor-deficient CHO mutant cell line) and PR230 (CMG2) (PR230 reconstituted with CMG2 receptor) were incubated with/without ET (0.1 or 1.0 μg/ mL) as indicated for 2 h. The ET-induced spherical morphological change was observed on PR230 (CMG2) cells but not on PR230 cells (A). Similarly, the ET-induced extracellular cAMP accumulation was only detected in the conditioned medium of PR230 (CMG2) cells but not that of PR230 cells (B). Means ± SD of three independent biological replicates. **C**. Co-culture experiment indicates no cell rounding effect of extracellular cAMP. The cover slips grown with PR230 cells and PR230 (CMG2) cells were placed in the same wells of a 6-well plate and co-incubated with/ without ET for 2 h. Of note, ET could only induce the characteristic cell morphological change to PR230 (CMG2) cells but not to PR230 cells in the co-culture system.

EF is an extremely efficient bacterial adenylate cyclase, able to convert 1000–2000 molecules of ATP to cAMP per second [36]. Therefore, EF may also deplete cellular ATP, thereby affecting a variety of ATP-depending biological processes, such as vesicle trafficking, tight junction protein delivery. To test whether the ET-induced cellular cAMP increases are accompanied by cellular ATP decreases, NIH3T3 cells were incubated with/without ET for various lengths of time. Remarkably, coincident with the increase in intracellular and extracellular cAMP, we noted a coincident decrease in intracellular ATP levels (Fig 4A–4C). Within 1 hour, cellular ATP had decreased by nearly 80%, followed by a transient elevation (likely due to compensatory increase in ATP production) and then a continuous decline (Fig 4C). Intriguingly, ET was also cytotoxic to the cells, and this was even more pronounced than LT-induced cytotoxicity (Fig 4D). To better understand the mechanism of the cytotoxicity of ET vs. LT, we did annexin V plus propidium iodide (PI) staining of the toxin-treated NIH3T3 cells, followed by flow cytometric analysis of the annexin V positive (early apoptotic cells) and PI positive (late apoptotic cells) cells (S3 Fig). While LT did not significantly kill cells, ET treatment resulted in approximately 50% undergoing apoptosis at 24 h and nearly 70% by 48 h (S3 Fig).

We next performed ET's cytotoxicity analysis on a panel of cells including human breast carcinoma MDA-MB-231 cells, mouse melanoma B16-F10 cells, mouse Glioma 261 cells, mouse primary fibroblasts, and CHO cells, and directly compared the cytotoxicity of ET and LT on these cells. Notably, ET exhibited more pronounced cytotoxicity than LT to all these

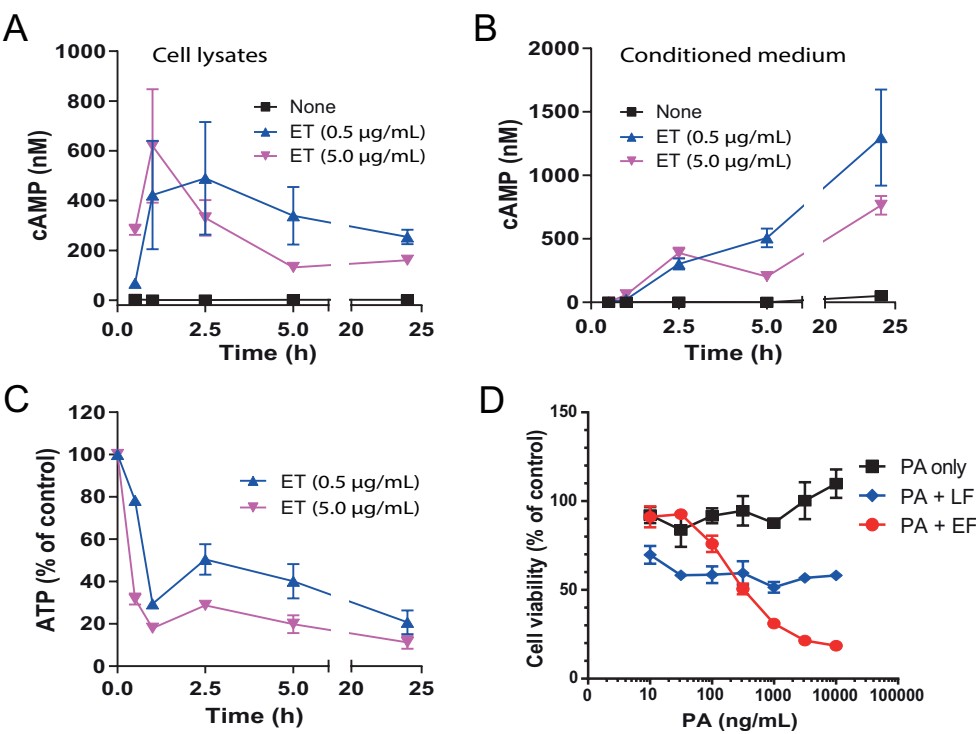

**Fig 4. ET-induced cAMP production, ATP depletion, and cytotoxicity to NIH3T3 cells. A-C.** The cells were incubated with/without ET for various lengths of time, followed by the measurements of cellular (A) and extracellular (B) cAMP levels, along with the cellular ATP levels (C). Means ± SD of three independent biological replicates. **D.** Cytotoxicity of ET and LT to NIH3T3 cells. Cells were incubated with various concentrations of PA in the presence of EF (1 μg/mL) or LF (1 μg/mL) for 48 h, followed by an MTT assay to assess cell viability. Means ± SD of three independent biological replicates.

cells (Fig 5A–5E). In parallel, ET rapidly reduced cellular ATP levels (in a dose-dependent manner) in similar patterns as described for NIH3T3 cells (Fig 4C), whereas LT only exhibited minimal effects (Fig 5F–5I). The ET-induced reduction in ATP levels was also verified in an *in vivo* study; as we noted hepatic ATP levels of ET-challenged mice were significantly decreased (Fig 5J).

## Cellular ATP depletion by ET but not by cholera toxin

Cholera toxin is another bacterial toxin that indirectly increases cellular cAMP through activating the endogenous adenylyl cyclases. Thus, the cAMP-increasing ability of cholera toxin would be limited by the availability of endogenous adenylyl cyclases. As such, in contrast to ET, which itself is a potent adenylate cyclase, cholera toxin could only gradually raise cAMP levels, in much slower pace and lesser extent than did ET (Fig 6A and 6B). As expected, while ET efficiently depleted cellular ATP, cholera toxin could not do so but instead induced up-regulation of ATP levels likely via a compensatory mechanism. Interestingly, forskolin, which also indirectly increases cAMP levels through activating endogenous adenylyl cyclases, could only temporarily decrease cellular ATP, following by compensatory up-regulation of ATP production (Fig 6C).

Taken together, the ET-induced ATP depletion appears to be an important mechanism underlying ET pathogenesis.

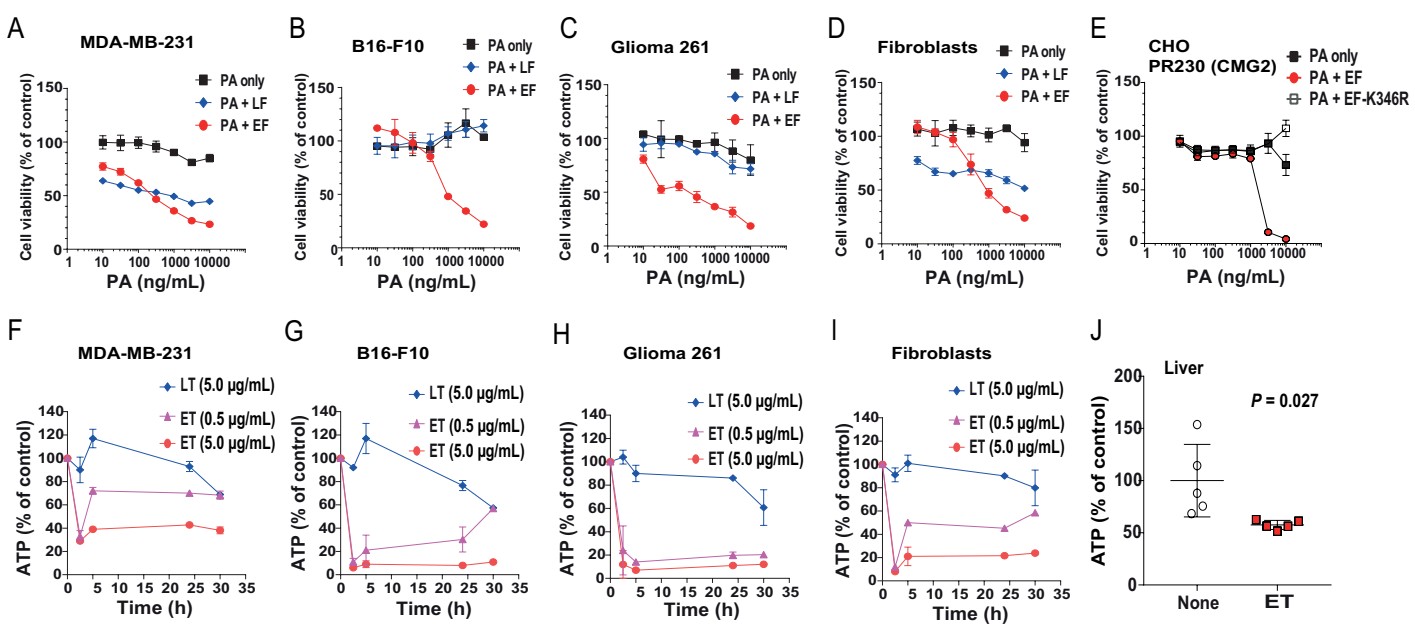

**Fig 5. ET-induced cytotoxicity and cellular ATP depletion to multiple cell lines. A-D**. Cytotoxicity of ET and LT to multiple cell lines. MDA-MB-231 (A), B16-F10 (B), Glioma 261 (**C**) cells, and mouse primary fibroblasts (**D**) were incubated with various concentrations of PA in the presence of EF (1 μg/mL) or LF (1 μg/mL) for 48 h, followed by an MTT assay to assess the cell viability. **E**. Cytotoxicity of ET to PR230 (CMG2) cells. The cells were incubated with various concentrations of PA in the presence of EF (1 μg/mL) or LF (1 μg/mL) for 48 h, followed by an MTT assay to assess the cell viability. **F-I**. Cellular ATP depletion by ET vs. LT in multiple cell lines. MDA-MB-231 (**F**), B16-F10 (**G**), Glioma 261 (**H**) cells, and mouse primary fibroblasts (**I**) were incubated with ET (0.5 μg/mL or 5 μg/mL) or LT (5 μg/mL) for various lengths of time as indicated, followed by measuring cellular ATP levels. (A-I) Means ± SD of three independent biological replicates. **J**. The significant reduction in cellular ATP levels in livers from the mice treated with ET. Eighteen hours after challenge with 30 μg ET (I.V.), the mice were euthanized, and livers collected and processed for measuring tissue ATP levels. Means ± SD. Unpaired two-tailed Student's *t*-test.

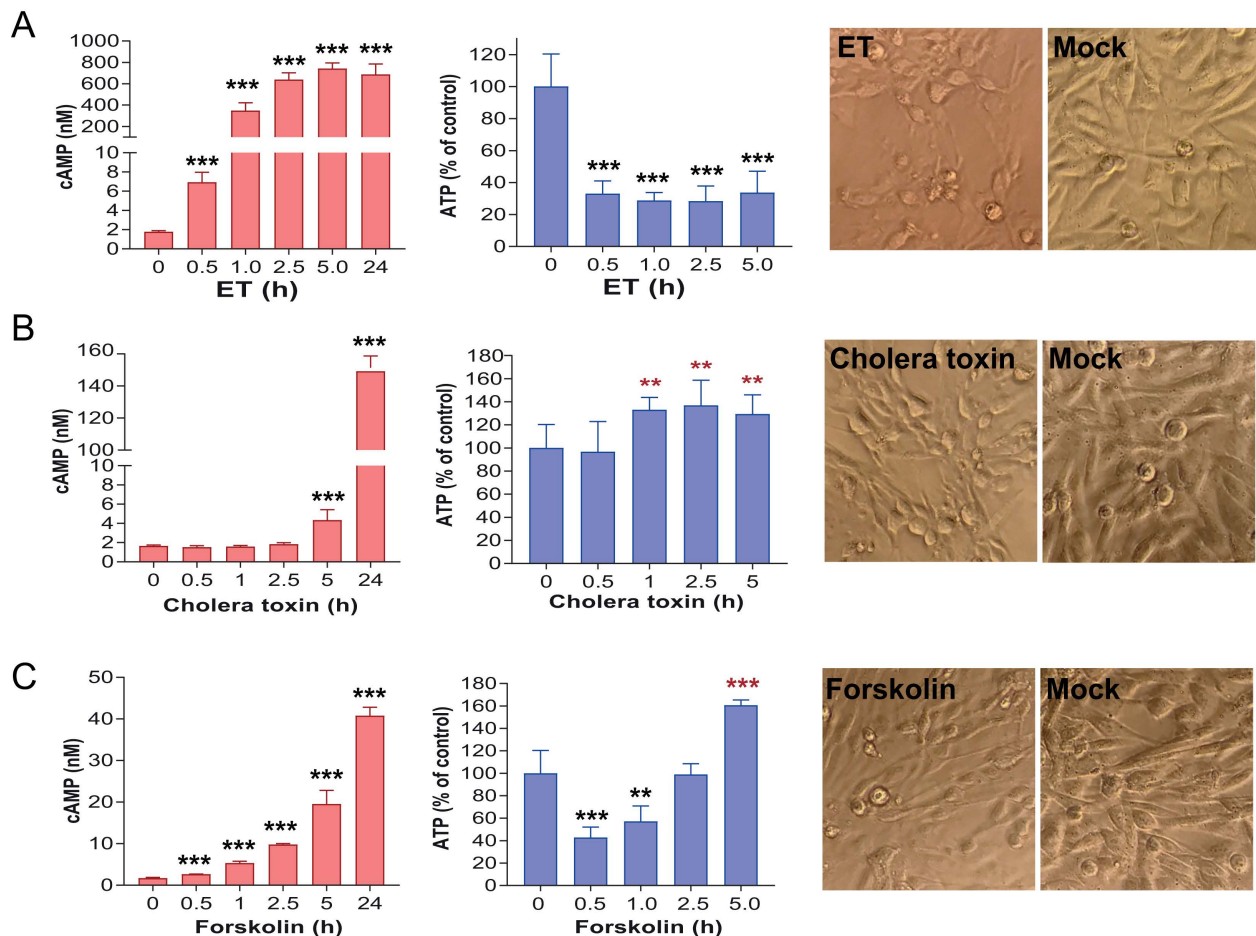

**Fig 6. ATP depletion by ET but not cholera toxin. A.** CHO cells were incubated with ET (PA/EF, each 2.5 µg/mL) for various lengths of time as indicated. Then cAMP (from conditioned medium, left panel) and cellular ATP (from cell lysates, middle panel) levels were assessed. Right panel, cell rounding morphological changes (at 24 h) were shown. ***, $p < 0.01$, when compared to the "0" groups. Unpaired two-tailed Student's *t*-test. **B**, **C**. CHO cells were incubated with cholera toxin (5 µg/mL) (**B**) or forskolin (12.5 µM) (**C**) for various lengths of time as indicated. Then cAMP (from conditioned medium, left panels) and cellular ATP (from cell lysates, middle panels) levels were assessed. Right panels, cell rounding morphological changes (at 24 h) were shown. **, $p < 0.05$, ***, $p < 0.01$, when compared to the "0" groups. Unpaired two-tailed Student's *t*-test. (A-C) Means ± SD of *t*hree independent biological replicates.

## Discussion

Certain protein toxins produced by diverse bacterial pathogens dramatically increase intracellular concentrations of cAMP, suggesting that over-production of this second messenger represents a strategy for bacterial pathogenesis. These toxins include adenylate cyclase toxins, such as anthrax edema toxin, *Bordetella pertussis* CyaA, and *Pseudomonas aeruginosa* ExoY, which rapidly convert ATP to cAMP [4,5,37,38]. Although cholera toxin, secreted by *Vibrio cholerae*, is not an adenylate cyclase, it can ADP-ribosylate the α subunit (Gαs) of trimeric G proteins, and locks Gαs in its GTP-bound, activated form, allowing Gαs to bind and activate endogenous adenylyl cyclases to produce cAMP [39]. Because cholera toxin is the direct mediator of the severe diarrheal symptoms seen in life-threatening cholera, the pathogenic mechanism of this toxin have been well studied [25]. It is widely believed that cholera toxin triggers cAMP-dependent, PKA activation in intestinal epithelial cells. The active PKA then phosphorylates the CFTR chloride channel protein on the surface of intestinal epithelial cells,

leading to ATP-mediated efflux of chloride ions, accompanied by secretion of $H_2O$, $Na^+$, $K^+$, and $HCO_3^-$ into the intestinal lumen. This causes rapid fluid loss from the intestine, resulting in severe dehydration of cholera patients that can culminate in death. As such, a small molecule inhibitor of CFTR was shown to be able to protect mice from acute diarrhea caused by toxigenic *V. cholerae* [40], attesting to the importance of CFTR in cholera toxin's pathogenesis.

Since anthrax ET is an adenylate cyclase and can cause intestinal fluid accumulation and tissue edema [26,29], it has been postulated that the cAMP-PKA-CFTR pathway may also mediate the *in vivo* toxicity of ET [4,5]. Here, indeed, we showed that ET can rapidly increase cellular cAMP and activate PKA. However, although the PKA inhibitor H89 could efficiently block PKA activation, surprisingly, we found that H89 could not protect mice from ET-induced mortality following a systemic challenge. Consistent with these observations, CFTR-null mice were found to be equally susceptible as WT control mice to ET challenges. Further, the potent CFTR inhibitor (CFTRInh-172) also could not protect mice from ET-induced mortality. Therefore, we conclude that ET-induced *in vivo* toxicity appears to be independent from the cAMP-PKA-CFTR pathway.

Unlike cholera toxin, EF itself is a direct and extremely efficient adenylate cyclase, able to convert 1000–2000 molecules/second of ATP to cAMP [36]. Therefore, in addition to increasing cellular cAMP, we found EF can also rapidly deplete cellular ATP, thereby potentially affecting a variety of biological processes dependent on ATP, such as vesicle trafficking (such as cell junction protein delivery) [41,42]. Unlike ET, cholera toxin increases cAMP through activation of the endogenous adenylyl cyclases with their activities regulated through negative feedback loops [43]. Therefore, the capacity of cholera toxin to convert ATP to cAMP is limited by the availability of endogenous adenylyl cyclases, and that cholera toxin is unlikely to be able to deplete ATP. In fact, CFTR's action is also ATP-dependent [25]. It is estimated that vesicle trafficking and active transport processes would consume approximately 50% of cellular ATP [41,42]. One characteristic cytotoxic effect of ET to cultured cells is its ability to rapidly induce the cell rounding morphological change, suggesting that the integrity of cell junctions is compromised in ET-treated cells. This is consistent with a study indicated that in the larval wing of *Drosophila melanogaster*, Rab11-mediated endocytic recycling was disrupted following exogenous expression of EF, resulting in cargo proteins such as cadherins failing to reach intercellular junctions [44]. Therefore, the ET-induced cell rounding, compromised cell junctions, and tissue edema are more likely due to the cellular ATP depletion rather than the cAMP over-production and the cAMP-mediated events. In fact, numerous studies have demonstrated the positive role of induction of cAMP in establishment and stabilization of intercellular junctions, through both the cAMP-activated PKA and Epac pathways [22,30,31,45,46]. Since sufficient levels of cellular ATP are key to various biological processes, here, we did consistently observe more pronounced cytotoxic effects of ET than LT to the cultured cells. Recently, we found that through downregulating the MEK-ERK-cMyc-bioenergetics axis, LT also has inhibitory effects on the cellular ATP production [47]. Therefore, *B. anthracis* has evolved to use these two exotoxins to synergistically reduce ATP levels via ATP depletion (ET) as well as its production (LT). Indeed, anthrax LT and ET demonstrate potent synergistic *in vivo* toxicity in challenged experimental animals [48,49].

In summary, our results support the notion that the ET-induced cellular ATP depletion rather than the cAMP overproduction contributes to ET lethality. Our results further suggest that ATP depletion might be an important mechanism underlying pathogenesis of other related adenylate cyclase toxins. However, the present work used a reductionist approach to investigate the role of ET in anthrax pathogenesis, and the results need to be further confirmed in studies using virulent *B. anthracis* strains.

## Methods and materials

### Ethics statement

All studies were caried out in with the protocols approved by the Institutional Biosafety Committee (IBC 201800198), the Institutional Animal Care and Use Committee of the University of Pittsburgh (#22030855), and by the National Institute of Allergy and Infectious Diseases, NIH (LPD 1E).

### Proteins and reagents

Recombinant PA, EF (with original N-terminal sequence MNE…), and LF (with original N-terminal sequence AGG…) proteins were purified from supernatants of BH500, an avirulent, sporulation-defective, protease-deficient *B. anthracis* strain, as described previously [50,51]. Cholera toxin (from *Vibrio cholera*, C8052-.5MG) was obtained from Sigma. H89 (S1582), CFTRInh-172 (S7139), forskolin (S2449), and ESI-09 (S7499) were purchased from Selleckchem. MTT (3-[4,5-dimethylthiazol-2-yl]-2,5-diphenyltetrazolium bromide, 475989-1GM) was obtained from Sigma.

### Cells and cytotoxicity assay

All cultured cells were grown at 37°C in a 5% $CO_2$ atmosphere. NIH3T3, B16F10, MDA-MB-231, and Glioma 261 cells were from ATCC, and were cultured in DMEM supplemented with 10% FBS. CHO PR230 and PR230 (CMG2) cells were described in [52], and were cultured in AMEM supplemented with 10% FBS. Primary fibroblasts were isolated from skins of newborn mice as described [53], cultured in DMEM supplemented with 10% FBS. For cytotoxicity assays, cells grown in 96-well plates were treated with various concentrations of the toxins as indicated for 48 h. Cell viabilities were then assessed by MTT as described previously [54], expressed as % of MTT signals of untreated cells. In co-culture experiments, PR230 cells and PR230 (CMG2) cells grown on different cover slips were placed into the same well of a 6-well plate and co-incubated with/without ET for 2 h. Cellular ATP levels were measured using ATPlite 1step kit (Perkin-Elmer, Boston, MA), following the manufacture's manual and are expressed as % of ATP levels of untreated cells. cAMP levels in cell lysates and in conditioned medium were measured with the HTRF cAMP HiRange Kit (Cisbio) according to the manufacturer's protocol.

For assessing the effect of H89 on PKA activation, NIH3T3 cells were serum-starved for 3 h. The cells were then incubated with ET (1 μg/mL) or forskolin (10 μM) with/without H89 (0.1 mM) for various lengths of time as indicated. Then cell lysates were prepared in the modified RIPA lysis buffer containing protease inhibitors as described [54]. Cell lysates were separated on SDS-PAGE gels, transferred onto nitrocellulose membranes, and analyzed by Western blotting using anti-phospho-CREB (Proteintech, 81871-1-RR) and anti-tubulin (Proteintech, 66031) antibodies.

For PI/annexin V staining, NIH3T3 cells treated with or without ET or LT were collected (including the detached cells in medium) and resuspended in 1× binding buffer (BD Biosciences) at a concentration of $1 \times 10^6$ cells/mL. Then, 0.1 mL of the solution was stained with 5 μL each of annexin V (BD Biosciences) and 50 μg/mL PI (Invitrogen) at room temperature for 15 min. The cells were analyzed using a BD FACSCanto Flow Cytometer and percentages of each cell population were obtained.

### Animal studies

*CFTR$^{-/-}$/Fabp2-CFTR* mice were from the Jackson Laboratory (Strain #: 002364). In ET challenge experiments, 8-10 week old male and female mice with various genotypes were injected

with ET (20-50 μg) in 0.5 ml PBS intraperitoneally or in 0.2 ml PBS intravenously. We used the following criteria to score mouse disease progression induced by ET: 0: healthy mouse; 1: slight ruffled coat but no problem in running around cage; 2: ruffled coat and decrease in activity; 3: ruffled coat, hunched posture and little movement; 3.5: moribund; 4: found dead. All toxin-challenged or infected mice were monitored twice daily for two weeks post challenge for signs of malaise or mortality. For testing PKA inhibitor H89, mice were administered with 30 μg ET (I.V.) or 30 μg ET (I.V.) plus 0.5 mg H89 (in 0.5 mL PBS, I.P.). For CFTR inhibitor CFTRinh-172 experiment, mice were administered (I.P.) twice at 24 h and 1 h with PBS or 50 μg CFTRinh-172 prior to ET injection (30 μg in 0.2 mL PBS, I.V.).

For ET footpad skin edema model, mice were injected intradermally in their right and left rear footpads with H89 (20 μg in 20 μl PBS) or CFTRinh-172 (12 μg in 20 μl 20% DMSO), and rear right footpads were injected with same volume of vehicles. Two hours later, the mice were injected intradermally in their right and left rear footpads with 0.2 μg ET or 0.2 μg CT in 20 μL PBS. The thicknesses of footpads (dorsal/plantar) were measured with a digital caliper before treatment, and at the indicated time points after ET or CT injection, expressed as % of footpad sizes before any injection.

To extract and measure cAMP levels of the tissues/organs from ET challenged mice, 18 h after challenge with 30 μg ET (I.V.), mice were euthanized, and blood, liver, heart, kidneys collected. cAMP levels from sera prepared from blood samples were measured with the HTRF cAMP HiRange Kit (Cisbio) according to the manufacturer's protocol. To prepare tissue extracts for cAMP and ATP measurements, we followed the procedure previously described [55]. Briefly, ~ 0.3 g each tissue sample was homogenized in 3 mL of ice-cold phenol-TE on ice. In 1 mL homogenate, 200 μL of chloroform and 150 μL of deionized water were added. The samples were then mixed well and centrifuged at 10,000 x g for 5 min, after which the upper aqueous phase (50 μL) was collected. The tissue extracts were then diluted for cAMP measurements using with the HTRF cAMP HiRange Kit (Cisbio). The liver extracts were also used to measure the ATP levels using ATPliteTM 1step (PerkinElmer, 6016731). ATP levels of the livers were normalized by the tissue wet weight, presented as percentages of heart ATP levels from the untreated mice.

## Statistics

Statistical significances of differences were calculated using unpaired two-tailed Student's *t*-test. Survival curves were compared using Log-rank test. *P* < 0.05 is considered as significant difference.

## Supporting information

**S1 Fig. Epac-independent in vivo toxicity of anthrax edema toxin.** A. Mice were injected with Epac inhibitor ESI-09 (200 μg, I.P.) 2 h prior to challenge with 50 μg ET (I.P.), and monitored for survival as indicated. Log-rank test for survival comparison. **B**. Mice were injected with Epac inhibitor ESI-09 (200 μg, I.P.) 2 h prior to injection with 0.25 μg ET (in 20 μl PBS) into their rear footpads. The thicknesses (dorsal/plantar) of footpads were measured with a digital caliper at the time points indicated after ET injection. Edema was expressed as % of footpad sizes before ET injection. Of note, ESI-09 could not diminish the ET-induced *in vivo* toxicity. Means ± SD. Unpaired two-tailed Student's t-test.
(EPS)

**S2 Fig. Amelioration of ET- and CT-induced footpad edema by PAK or CFTR inhibition.** A, B. ET-induced footpad edema in mice. Mice were injected with 0.2 μg ET (in 20 μl PBS) into their rear right and left footpads, two hours after the rear right footpads were pre-injected

with H89 (20 μg in 20 μl PBS) (C) or CFTRinh-172 (12 μg in 20 μl 20% DMSO) (D), with their rear left footpads were injected with same volume of vehicle. The thicknesses of footpads were measured with a digital caliper before injection and at the time points indicated after ET injection, expressed as % of footpad sizes before injection. **C, D**. CT-induced footpad edema in mice. Mice were injected with 0.2 μg CT (in 20 μl PBS) into their rear right and left footpads two hours after their rear right footpads were injected with H89 (20 μg in 20 μl PBS) (E) or CFTRinh-172 (12 μg in 20 μl 20% DMSO) (F), and their left rear footpads were injected with same volume of vehicle. The thicknesses of footpads were measured before injection and at the indicated time points after CT injection, expressed as % of footpad sizes before injection. Of note, H89 and CFTRinh-172 could ameliorate the CT-induced footpad edema. (A-D) Means ± SD of independent biological repeats (each symbol represents one mouse). Unpaired two-tailed Student's *t*-test.
(EPS)

**S3 Fig. Apoptotic cell death induced by ET.** A. NIH 3T3 cells were incubated with or without ET (5 μg/mL), or LT (5 μg/mL) for various lengths of time, followed by flow cytometry analyses of the cells after staining with propidium iodide (PI) and annexin V. B. Images of the NIH 3T3 cells treated with/without of ET (5 μg/mL), LT (5 μg/mL), or PA + FP59 (0.1 μg/mL PA + 0.1 μg/mL FP59) for 48 h.
(EPS)

## Author contributions

**Conceptualization:** Jie Liu, Shihui Liu.

**Formal analysis:** Shihui Liu.

**Funding acquisition:** Shihui Liu.

**Investigation:** Jie Liu, Qing Cao, Michael Ewing, Zehua Zuo, Shihui Liu.

**Methodology:** Jie Liu, Qing Cao, Zehua Zuo, Jason R. Kennerdell.

**Project administration:** Jie Liu, Shihui Liu.

**Resources:** Jie Liu, Jason R. Kennerdell, Toren Finkel, Stephen H. Leppla.

**Supervision:** Jie Liu, Shihui Liu.

**Validation:** Jie Liu, Qing Cao, Shihui Liu.

**Writing – original draft:** Jie Liu, Shihui Liu.

**Writing – review & editing:** Toren Finkel, Stephen H. Leppla.

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
