## [Decision Letter · Decision Letter 0]

4 Mar 2025

Dear Dr. Liu,

We are pleased to inform you that your manuscript 'ATP depletion in anthrax edema toxin pathogenesis' has been provisionally accepted for publication in PLOS Pathogens.

Best regards,

Joseph A Sorg, Ph.D.

Academic Editor

PLOS Pathogens

Helena Boshoff

Section Editor

PLOS Pathogens

Sumita Bhaduri-McIntosh

Editor-in-Chief

PLOS Pathogens

orcid.org/0000-0003-2946-9497

Michael Malim

Editor-in-Chief

PLOS Pathogens

orcid.org/0000-0002-7699-2064

Reviewer Comments (if any, and for reference):

Reviewer's Responses to Questions

**Part I - Summary**

Reviewer #1: (No Response)

Reviewer #2: Anthrax edema toxin has long been held as a toxin that cause swelling/edema, but it turned out purified edema toxin is quite toxic and lethal, even more so that anthrax lethal toxin in certain animal models. The authors here show that edema toxin may exert this effect by depleting cellular ATP by its rapid adenylate cyclase activity. This is a new and exciting direction for this research field. The authors of course could have done some work with pathogenic Bacillus anthracis to further support their findings with purified toxins, but these experiments are probably suited better for future research.

Reviewer #3: My concerns have been properly addressed.

**Part II – Major Issues: Key Experiments Required for Acceptance**

Reviewer #1: (No Response)

Reviewer #2: I have no major concerns. The authors address my issues as well as the other two reviewers' concerns.

Reviewer #3: None.

**Part III – Minor Issues: Editorial and Data Presentation Modifications**

Reviewer #1: (No Response)

Reviewer #2: None.

Reviewer #3: None.

PLOS authors have the option to publish the peer review history of their article (what does this mean? ). If published, this will include your full peer review and any attached files.

**Do you want your identity to be public for this peer review?** For information about this choice, including consent withdrawal, please see our Privacy Policy .

Reviewer #1: No

Reviewer #2: No

Reviewer #3: No

---

## [Editor Report · Acceptance letter]

Dear Dr. Liu,

We are delighted to inform you that your manuscript, "ATP depletion in anthrax edema toxin pathogenesis," has been formally accepted for publication in PLOS Pathogens.

Best regards,

Sumita Bhaduri-McIntosh

Editor-in-Chief

PLOS Pathogens

orcid.org/0000-0003-2946-9497

Michael Malim

Editor-in-Chief

PLOS Pathogens

orcid.org/0000-0002-7699-2064